# Enhancing Acetate Utilization in *Phaeodactylum tricornutum* through the Introduction of Acetate Transport Protein

**DOI:** 10.3390/biom14070822

**Published:** 2024-07-09

**Authors:** Pu Song, Ning Ma, Shaokun Dong, Hongjin Qiao, Jumei Zhang, Bo Guan, Shanying Tong, Yancui Zhao

**Affiliations:** 1School of Life Sciences, Ludong University, Yantai 264025, China; 2021110248@m.ldu.edu.cn (P.S.); 2022110265@m.ldu.edu.cn (N.M.); 2022110268@m.ldu.edu.cn (S.D.); jmzhang@ldu.edu.cn (J.Z.); 3500@ldu.edu.cn (S.T.); yancuizh@ldu.edu.cn (Y.Z.); 2School of Resources and Environmental Engineering, Ludong University, Yantai 264025, China; bguan@ldu.edu.cn

**Keywords:** *Phaeodactylum tricornutum*, acetate, acetate transport protein, photosynthetic activities, fatty acid composition

## Abstract

The diatom *Phaeodactylum tricornutum*, known for its high triacylglycerol (TAG) content and significant levels of n-3 long chain polyunsaturated fatty acids (LC-PUFAs), such as eicosapentaenoic acid (EPA), has a limited ability to utilize exogenous organic matter. This study investigates the enhancement of acetate utilization in *P. tricornutum* by introducing an exogenous acetate transport protein. The acetate transporter gene *ADY2* from *Saccharomyces cerevisiae* endowed the organism with the capability to assimilate acetate and accelerating its growth. The transformants exhibited superior growth rates at an optimal NaAc concentration of 0.01 M, with a 1.7- to 2.0-fold increase compared to the wild-type. The analysis of pigments and photosynthetic activities demonstrated a decline in photosynthetic efficiency and maximum electron transport rate. This decline is speculated to result from the over-reduction of the electron transport components between photosystems due to acetate utilization. Furthermore, the study assessed the impact of acetate on the crude lipid content and fatty acid composition, revealing an increase in the crude lipid content and alterations in fatty acid profiles, particularly an increase in C16:1n-7 at the expense of EPA and a decrease in the unsaturation index. The findings provide insights into guiding the biomass and biologically active products production of *P. tricornutum* through metabolic engineering.

## 1. Introduction

Oleaginous microalgae have become a focal point for the development of sustainable biofuel and bioproducts. Particularly, polyunsaturated fatty acids (PUFAs) are of high interest due to their recognized health advantages for human consumption [1]. As a model diatom species, *Phaeodactylum tricornutum* contains high content of triacylglycerols (TAGs) and significant amounts of the n-3 long chain PUFA-like eicosapentaenoic acid (EPA) [2,3]. This organism is capable of photoautotrophic growth, utilizing carbon dioxide (CO_2_) as its carbon source, which is predominantly accumulated as lipid, constituting up to 57.8% of dry cell weight [4]. Furthermore, mixotrophic or heterotrophic growth has also been widely reported with glycerol, glucose, and acetate as the main carbon sources [5,6,7]. However, the actual utilization capacity of organic carbon sources by *P. tricornutum* varies significantly due to differences in algal strains and experimental conditions.

In the past, *P. tricornutum* had been regarded as a strictly autotrophic organism, with a limited ability to utilize exogenous organic matter [7,8]. However, a variety of organic substances, including saccharides, alcohols, carboxylic acids, amino acids, and phenolic compounds, have been found to be utilizable by *P. tricornutum* [9]. *P. tricornutum* can utilize glycerol, acetate, and glucose at concentrations up to 100 mM, achieving growth rates 1.60, 1.28, and 1.21 times higher than autotrophic conditions, respectively [5]. Nonetheless, this is less than the heterotrophic growth of species like *Chlorella*, which can surpass autotrophic growth by more than twofold and even by an order of magnitude [10]. Similar to the heterotrophic growth of green algae, the heterotrophic growth of *P. tricornutum* also has a significant impact on its photosynthesis and biochemical composition. Heterotrophic growth leads to a reduction in the content of photosynthetic pigments, and a decrease in the efficiency of photosynthesis [5,11]. Lipid content is usually increased and the composition of fatty acids is altered, with an increase in the levels of saturated and monounsaturated fatty acids and a decrease in the levels of PUFAs [12].

Glucose and acetate have been widely reported as two main carbon sources that microalgae can utilize. Microalgae that can process organic materials are sorted into two groups: those that mainly use acetate, and those that mostly use sugars [13]. *P. tricornutum* is capable of mixotrophic growth utilizing acetate and glucose as carbon sources, although its growth rate is markedly lower than when utilizing glycerol as a carbon source. Glycerol can freely diffuse into the cells, whereas acetate and glucose typically require active transport mechanisms to enter the cells. It is likely that *P. tricornutum* lacks the necessary transporters to facilitate the uptake of acetate and glucose. This hypothesis is substantiated by the observation that the introduction of glucose transporter protein into *P. tricornutum* has enabled the organism to utilize glucose, resulting in a growth rate that significantly surpasses that of the wild-type strain [14,15]. Additionally, this has also led to an increase in lipid content and alterations in the composition of fatty acids, as well as changes in the pathways of glucose metabolism. However, there have been no reports concerning acetate transporter in *P. tricornutum* to date. Acetate presents significant potential as a carbon source for mixotrophic cultivation. It has several benefits over glucose, including being more economical and readily accessible. Acetate can be produced from C1 gases such as CH_4_, CO, and CO_2_ [16]. Moreover, acetate’s antibacterial properties reduce the likelihood of contamination [17]. It also promotes the synthesis of biomass and lipids by intensifying the carbon flow through the tricarboxylic acid cycle by providing the acetyl coenzyme A as substrate [18].

This study aims to investigate whether the assimilation of acetate by *P. tricornutum* requires the presence of an acetate transporter protein and to explore whether the introduction of such a protein would alter its photosynthetic activity and lipid metabolism. The findings of this research could provide guidance for the biomass production of *P. tricornutum* and its application in the production of biologically active products such as PUFAs.

## 2. Materials and Methods

### 2.1. Microorganisms and Culture Media

*P. tricornutum* CCAP 1055/1 was acquired from the Culture Collection of Algae and Protozoa (CCAP). The cultivation was carried out in artificial seawater prepared with silicon-free f/2 medium [19]. When carrying out the mixotrophic growth, sodium acetate (NaAc) was added to f/2 medium.

*Saccharomyces cerevisiae* was cultivated in the yeast extracts-peptone-dextrose (YPD) medium at 28 °C with shaking at 150 rpm. It was used to obtain acetate transporter gene *ADY2* (Gene ID: 850368) [20].

### 2.2. ADY2 Modification of P. tricornutum

The acetate transporter gene *ADY2* was obtained through PCR amplification of *S. cerevisiae* genome with primers listed in Table 1. *ADY2* was ligated to the downstream of fcpA promotor in the plasmid pPha-T1 [21] (Appendix A) by T4 DNA ligase after digestion with Bam HI and Sph I (Sangon Biotech. Co., Ltd., Shanghai, China). The ligation product was transformed into *Escherichia coli* TOP10 cells (Sangon Biotech. Co., Ltd.) and incubated at 37 °C overnight. After incubation, it was spread onto an LB solid medium and cultured for 12 h. Positive clones were screened using colony PCR technology and sent to Sangon Biotech. Co., Ltd. for sequencing.

Positive clone strains were expanded using LB culture and the plasmids were extracted. The recombinant plasmids were then bombarded into *P. tricornutum* as described by Krämer et al. [22]. The cells were cultured at 21 °C with a light intensity of 60 μmol m^−2^ s^−1^ for 24 h and then washed with an equal volume of PBS and transferred into a 1.5 mL sterile centrifuge tube. Subsequently, they were spread onto a solid f/2 medium containing 1 mg mL^−1^ bleomycin (Invitrogen) and cultured at 21 °C, a light intensity of 60 μmol m^−2^ s^−1^, and a light:dark cycle of 12 h:12 h. After 15 days of cultivation, single colonies appeared. The single colonies were picked and expanded in a 12 mL culture tube containing 10 mL f/2 medium with 0.7 mg mL^−1^ bleomycin. Three transformants (ADY2-4, ADY2-9, and ADY2-12) and the wild-type *P. tricornutum* were inoculated into fresh medium for the subsequent growth evaluation.

### 2.3. Culture and Growth

To screen for the transformant that grows rapidly under the condition of NaAc supplementation and to determine the optimal concentration of NaAc for growth, three transformants and the wild-type strain were each cultured in f/2 medium containing NaAc at concentrations of 0, 0.005, 0.01, 0.05, and 0.1 M for 7 days to compare the specific growth rates. The growth curves under NaAc concentrations of 0 M and 0.01 M were plotted to assess the growth differences between the transgenic and wild-type strains. After selecting the rapid growth strain, a comparative analysis was conducted on the dry weight content of the mutant strain and the wild-type strain after 7 days of growth under 0 M and 0.01 M NaAc conditions. Additionally, the specific growth rates of the mutant and wild-type strains were compared and analyzed after 7 days of growth under conditions of 0.01 M and 0.05 M sodium propionate and sodium butyrate. The inoculation and cultivation procedures were carried out according to the following steps.

Algal cultures in the logarithmic growth phase were collected via centrifugation at 4000× *g* for 10 min and rinsed twice with artificial seawater. The resulting pellets were subsequently suspended in 500 mL of medium to create the inoculum. Cells were cultivated at a temperature of 21 °C under cool white fluorescent lamps (100 μmol m^−2^ s^−1^) with a light:dark cycle of 12 h for 7 days, and the cultivation was conducted with shaking at a speed of 150 rpm. Each bottle of algal culture was inoculated with an initial density of 10^6^ cells mL^−1^. Daily sampling for analysis was conducted 1 h after the end of the light cycle. Cell density measurements were performed using a cell counter (Ruiyu, Shanghai, China), and the specific growth rate (*μ*) was calculated using the following formula.
*μ* = (*lnX_t_* − *lnX*_0_) (*t_x_* − *t*_0_)^−1^(1)
where *X_t_* and *X*_0_ represent the cell density at culture time *t_x_* and *t*_0_, respectively.

After measuring the cell density on the 7th day, the algal cells were collected by centrifugation at 4000× *g* for 10 min and washed twice with PBS solution before proceeding with the subsequent experiments.

### 2.4. Dry Weight

An accurate volume of the culture was filtered onto a pre-dried and pre-weighed glass fiber filter (with a pore size of 0.22 μm) using a vacuum pump. Then, the filter was washed with a 0.5 M ammonium formate solution to remove salts. The same volume of filtered seawater was filtered as a control following the same procedure. Both filters were dried at 100 °C for 4 h to allow the ammonium formate to evaporate and then weighed on an analytical balance, respectively. We calculated the dry weight per liter culture according to Formula (2):DW (g·L^−1^) = (DW_A_ − DW_C_)·V^−1^(2)
where DW_A_ is the average dry weight retained on algal filter (g), DW_C_ is the average dry weight retained on control filter (g), and V is the volume of algal culture filtered on the filter (L).

### 2.5. Photosynthetic Pigments

Five milliliters of the algal culture cultivated up to the 7th day were filtered through a GF/F membrane (Whatman, Maidstone, UK). The membrane was then placed into a centrifuge tube in an ice bath. Thereafter, 5 mL of 90% acetone were immediately added to the centrifuge tube, which was subsequently wrapped in aluminum foil to avoid light. The tube was shaken and allowed to stand in a 4 °C refrigerator for 24 h, with intermittent shaking at intervals of every 3 to 4 h. Following this, the mixture was centrifuged at 10,000× g for 10 min at 4 °C. The resulting supernatant was carefully transferred to a fresh centrifuge tube.

The absorbance of the supernatant was then measured using a spectrophotometer at specific wavelengths, namely 750 nm, 663 nm, 645 nm, 630 nm, and 470 nm. The absorbance values obtained at each of these wavelengths were corrected by subtracting the absorbance measured at 750 nm. These corrected absorbance values were subsequently divided by the cuvette path length, which was 1 cm, to yield the respective values of D663, D645, D630, and D470. Finally, the concentrations of chlorophyll a (*Chl-a*) and *c* (*Chl-c*) and carotenoids present in the extract were calculated using the following formula [23]:*Chl-a* (μg mL^−1^) = 11.64 × D_663_ − 2.16 × D_645_ + 0.10 × D_630_,(3)
*Chl-c* (μg mL^−1^) = −5.53 × D_663_ − 14.81 × D_645_ + 54.22 × D_630_,(4)
Carotenoid (μg mL^−1^) = (1000 × D_470_ − 3.27 × *Chl-a*)/229(5)

We calculated the content of each pigment inside the cells according to the following formula, with units expressed in micrograms per 10^7^ cells.
Pigment (μg 10^−7^ cells) = (C(pigment) × V_Acetone_)/(D_cell_ × V_sample_) × 10^7^(6)
where C(pigment) represents the concentration of each photosynthetic pigment in the extract, V_Acetone_ is the volume of acetone (mL), D_cell_ is the cell density (cells mL^−1^), and V_sample_ is the volume of the algal sample (mL).

### 2.6. Photosynthetic Activity

On the 7th day, the photosynthetic activity of 2 mL cultures from each group was measured using a fluorimeter (AquaPen AP 100, Photon Systems Instruments, Drasov, Czech Republic). Each sample was dark-adapted for 15 min, and then the maximal photochemical efficiency of photosystem II (*F_v_⁄F_m_*), the effective photochemical efficiency of PSII in the light [QY = yield = *F_v_*′/*F_m_*′ = (*F_m_*′ − *F_o_*′)/*F_m_*′], the OJIP test, and the light response curve protocol (LC3) were carried out following the instructions of the manufacturers.

### 2.7. Microscopic Observation

The wild-type and mutant cells in the late-exponential growth phase were stained with fluorescent neutral lipid stain Nile Red to detect the lipid droplets [24,25]. A solution of Nile Red (0.1 mg mL^−1^ in acetone) was combined with 20 μL of DMSO and 80 μL of cell culture. The resulting mixture was then stained for 30 min at room temperature in a dark environment, followed by observation using an Axio Observer fluorescence microscope (Carl Zeiss, Gottingen, Germany) equipped with a Colibri 7-illumination module. Fluorescence was excited with a blue light at 475 nm.

### 2.8. Crude Lipid Content

The crude lipid content was determined according to a modified procedure of Folch et al. [26]. Typically, 50–100 μL (approximately 20 mg) of freeze-dried cells was weighed into a centrifuge tube. Then, 1 mL of methanol and 2 mL of chloroform were added. The mixture was vigorously vortexed for 2 min on a vortex mixer, followed by centrifugation for 5 min at 5000× *g* and 4 °C. The supernatant was transferred to a new tube, and 1/5 volume (about 600 μL) of NaCl solution (0.9% *w*/*w*) was added. After vigorous vortexing for 2 min, the mixture was allowed to stand for 5 min until the layers separated. The upper layer was removed, and the lower layer containing the crude lipid extract was retained. After evaporating the solvent under a stream of nitrogen, the residue was dried at 85 °C for 15 min until reaching a stable weight. Then, it was placed in a desiccator or immediately weighed using a precision balance. The crude lipid content was calculated by the difference between the total mass of the centrifuge tube after the test and the mass of the empty centrifuge tube before the test.

### 2.9. Fatty Acid Composition

The fatty acid composition was analyzed according to our prior publication [27]. Typically, an appropriate amount of wet cell pellet (c.a. 100 mg) was accurately weighed into a screw-capped test tube (12 cm × 1.5 cm). Half a milliliter of toluene containing internal standard (0.1 mg glyceryl triheptadecanoate) and 1 mL of 0.5 M NaOH methanol solution were slowly added and vortex-mixed at 80 °C with constant stirring at 300 rpm for 20 min. The mixture was cooled to room temperature for 5 min, followed by the addition of 1 mL of 10:100 (*v*/*v*) acetyl chloride in anhydrous methanol, repeating the above incubation procedure. After cooling for 5 min to room temperature, 1 mL of 6% K_2_CO_3_ and 400 μL of n-hexane were added and vortex-mixed, and the supernatant was extracted by centrifugation at 4000× *g* for 1 min, followed by collection of the upper layer. The resulting fatty acid methyl esters were then subjected to analysis via gas chromatography (GC-1949; Panna, Changzhou, China) equipped with a fused silica capillary column (Supelco SP-2560: 100 m × 0.25 mm, film thickness 0.20 μm; Bellefonte, PA, USA). The heater was maintained at 260 °C with nitrogen as the carrier gas. The column temperature was ramped from 140 to 260 °C at a rate of 10 °C per minute. Fatty acids were identified by correlating the relative retention times with those of the reference standards (Merck, Rahway, NJ, USA). The unsaturation index (UI) of fatty acids was calculated using the following formula [28]:UI = 1 × (% monoenoic) + 2 × (% dienoic) + 3 × (% trienoic) + 4 × (% tetraenoic) + 5 × (% pentaenoic) + 6 × (% hexaenoic)(7)

### 2.10. Chemical Oxygen Demand (COD)

To assess the consumption rate of NaAc during growth, the chemical oxygen demand (COD) was measured before and after cultivation to represent the rate of NaAc consumption. Cultivated until the 7th day, 100 mL of algal culture was taken and centrifuged at 4000× *g* for 10 min. The supernatant and the initial media without inoculation were used for COD measurement. According to the national standard of China GB17378.4-2007 [29], COD was determined using the alkaline potassium permanganate method.

### 2.11. Statistical Analysis

Experimental results were expressed as the mean ± standard deviation (SD) of the three replicates. The Kolmogorov–Smirnov test and Levene’s F-test were used to check normality and heterogeneity of variances. Data with parametric distribution were subjected to one-way analysis of variance (more than two groups) or *t*-test (two groups). Duncan multiple-range test [30] or Least-Significant-Difference test [31] was performed to identify significant differences between any two means that differed at *p* < 0.05. All statistical analyses were performed using IBM SPSS statistics version 20.0 (IBM, Armonk, NY, USA).

## 3. Results and Discussion

### 3.1. Evaluation of Growth and NaAc Assimilation

The *ADY2* from *S. cerevisiae* was successfully inserted into the plasmid pPha-T1 and introduced into *P. tricornutum* (Appendix A). Three transformants (ADY2-4, ADY2-9, and ADY2-12) were obtained. By comparing the specific growth rates of the wild-type (WT) and transgenic strains at different NaAc concentrations for 7 days, we observed the optimal growth conditions for all strains at the NaAc concentration of 0.01 M (Figure 1). In low concentrations of NaAc (0–0.005 M), the growth of ADY2-4 and ADY2-9 was significantly superior to that of WT (*p* < 0.05), whereas there was no significant difference in growth between ADY2-12 and WT (*p* > 0.05). In contrast, at high concentrations of NaAc (0.01 M–0.1 M), the growth of all three mutant strains was significantly better than that of WT (*p* < 0.05), demonstrating that *ADY2* was transferred into the *P. tricornutum* cell and conferred it the ability of acetate assimilation to accelerate growth. The growth rate of the transformants was increased by 1.7–2.0 fold in comparison with WT at the NaAc concentration of 0.1 M. However, the maximum growth rate of all three mutant strains appeared at the NaAc concentration of 0.01 M. Wang et al. (2012) tested the capability of *P. tricornutum* to utilize NaAc within the concentration range of 0.5 g L^−1^ to 5 g L^−1^, and it was found that the algal cells exhibited optimal growth at a NaAc concentration of 0.5 g L^−1^ (0.006 M) [12]. Haro et al. [30] also reported that a NaAc concentration of 0.01 M promoted the best growth [32]. The results of these reports were consistent with the WT in this study; however, the mutant strains of ADY2-4 and ADY2-9 still grew well without affecting the growth rate at a NaAc concentration of 0.1 M. Actually, we found that the ADY2-4 strain ceased to grow under conditions of up to 1 M NaAc (Appendix A). To the best of our knowledge, there are currently no reports on the expression of acetate transport protein in *P. tricornutum*. The present study demonstrates that the expression of acetate transport protein in *P. tricornutum* can enhance the growth of the algae as effectively as the expression of glucose transport proteins [33,34].

Subsequently, we selected 0.01 M as the testing concentration of NaAc to investigate the growth process. As shown in Figure 2, all strains completed the logarithmic phase within 7 days, regardless of the conditions with or without the addition of NaAc. This trend is consistent with recent reports regarding the algal strain [35,36,37]. There was a noticeable increase in cell density during the period from the 2nd to the 6th day. Starting from the 3rd day, the cell densities of ADY2-4 and ADY2-9 were significantly higher than the other groups (F(7, 16)= 15.80, *p* < 0.05). By the 7th day of cultivation, this effect became even more pronounced, with the cell densities of ADY2-4 and ADY2-9 reaching 16.28 and 16.24 × 10^6^ cells mL^−1^, respectively. After comparison, we proceeded with the subsequent experiments using the ADY2-4 strain.

Without NaAc treatment, there was no significant difference in the dry weight between the wild-type and mutant strain (*p* > 0.05, Figure 3a). However, after NaAc treatment, the dry weight of the mutant strain ADY2-4 increased significantly (*p* < 0.05, Figure 3a). Compared to the control treatment WT-0M and WT-0.01M, the dry weight of ADY2-4-0.01M increased by 66.97% and 39.20%, respectively, indicating that the modification of acetate transport proteins has the potential to enhance the biomass of *P. tricornutum*.

To explore whether the mutant strain can grow normally in sodium propionate and sodium butyrate, we added 0.01 M and 0.05 M sodium propionate and sodium butyrate separately to f/2 medium and measured the specific growth rate of the wild-type and mutant strain. As shown in Figure 3b, under low concentrations of sodium propionate (0.01 M) and sodium butyrate (0.01 M) treatment, there was no significant difference in specific growth rate between the wild-type and mutant strain (*p* < 0.05). Under high concentrations of sodium propionate (0.05 M) and sodium butyrate (0.05 M) treatment, the growth of both strains was inhibited, while the growth of the wild-type was nearly halved in comparison to the mutant type. The results indicate that the ADY2-4 strain not only has the ability to utilize NaAc but also has the capability to assimilate sodium propionate and sodium butyrate. This suggests that the acetate transport protein exhibits a generality in the transport of small fatty acid molecules [38].

The NaAc assimilation capacity of algal cells can be reflected by the changes in COD values. As shown in Table 2, by the 7th day of cultivation, the utilization rate of NaAc by the wild-type was 86.23 ± 0.25%, while that of the mutant strain was 90.30 ± 0.27%, significantly higher than the wild-type (*p* < 0.05). It is not easy to determine the concentration of NaAc in seawater. Changes in the concentration of COD can effectively represent the variations of acetate concentration. Observing the COD removal rate, both the wild type and mutant strain are capable of utilizing NaAc in the culture medium. The mutant strain continues to absorb NaAc even at low concentrations, suggesting that the acetate transport protein may carry out the active transport at low acetate concentrations.

### 3.2. Photosynthesis Evaluation

#### 3.2.1. Photosynthetic Pigments

There was a significant difference in the pigment content between the wild-type and mutant strains (Figure 4). The content of each pigment in the ADY2-4 was higher than that in the wild type when culturing in the NaAc-free medium, with *Chl-a* increasing by 33.32%, *Chl-c* by 14.41%, and carotenoids by 31.45%. However, after NaAc treatment, the content of each pigment in the WT-0.01M and ADY2-4-0.01M was significantly lower than that in the NaAc-free groups (*p* < 0.05). It is noteworthy that the content of each pigment in the ADY2-4-0.01M was significantly lower than that in the WT-0.01M (*p* < 0.05), with *Chl-a* reducing by 35.46%, *Chl-c* by 30.73%, and carotenoids by 35.03%. Consistent results of pigments’ decline when treated with acetate in *P. tricornutum* were reported by Liu et al. [5]. Similar results were also obtained in other microalgae when cultivated with acetate or glucose [39,40,41]. It appears that the mutant strain had absorbed a greater amount of NaAc (Table 2), leading to a lower pigment content compared to the wild type (Figure 4). This observation may be attributed to acetate inhibiting the expression of pigment-related proteins [42]. Interestingly, the mutant demonstrated a higher pigment content in the absence of NaAc compared to the wild type. Further investigation is required to elucidate the mechanism underlying this phenomenon.

#### 3.2.2. Photosynthetic Activities

On the 7th day of treatment, there was no significant difference in *F_v_*/*F_m_* between the wild-type and ADY2-4 in the NaAc blank groups. However, after treatment with NaAc, the *F_v_*/*F_m_* of ADY2-4 decreased significantly compared to the blank groups (*p* < 0.05, Figure 5a). Regarding QY, in the blank groups, the mutant strain ADY2-4 had a slightly higher QY value than the wild-type. After treatment with NaAc, the QY values decreased in both strains, but there was no significant difference between the wild-type and mutant strain ADY2-4 (Figure 5b). Liu et al. reported that photosynthesis activities including *F_v_*/*F_m_*, CO_2_ fixation, carbohydrase activity, and net O_2_ release were inhibited by acetate in *P. tricornutum* [5]. Our previous report also found the apparent decline of *F_v_*/*F_m_* and QY in *Chlorella sorokiniana* when treated with acetate [43].

In order to study the effect of NaAc on PSII electron transfer reactions, we assessed the rapid chlorophyll fluorescence induction curves in the absence and presence of NaAc. Under both conditions, the chlorophyll fluorescence induction kinetics curves of the two algal strains exhibited distinct O, J, I, and P phases (Figure 5c). Under all treatment conditions, the trend of fluorescence value changes from point O to point J was consistent, indicating that the NaAc culture had no significant effect on the electron transfer from QA to QB of PSII. After the addition of NaAc, the fluorescence values of WT-0.01M and ADY2-4-0.01M increased rapidly from point J to point I, resulting in relatively variable fluorescence (VI) at point I being higher than the corresponding blank group, respectively. This suggests that the transfer of PSII electrons from QB to PQ pool was inhibited by NaAc culture. The light response curves of all treatments exhibited an initial increase followed by a gradual stabilization trend (Figure 5d). In the absence of NaAc, the mutant strain’s maximum electron transport rate (ETR_max_) was significantly higher than that of the control group (F(2, 2) = infinity, *p* < 0.05). This trend aligns with the pattern of changes observed in photosynthetic pigments (Figure 4). However, after the addition of NaAc, the ETR_max_ of both the wild-type and mutant strains decreased with no significant difference (F(2, 2) = 1, *p* > 0.05), which is consistent with the trend in the variation of *F_v_*/*F_m_* and QY (Figure 5a,b). These findings also align well with the report of Liu et al. [5].

Upon entry into the cell via an acetate transport protein, exogenous acetate undergoes conversion by acetyl coenzyme A synthetase to form acetyl coenzyme A with ATP consumption [44]. Acetyl coenzyme A enters the glyoxylate cycle, and ultimately generates oxaloacetic acid to drive the TCA cycle within the mitochondria or produce carbohydrates through gluconeogenesis. Therefore, acetate addition generally accompanies a drop in the ATP content and a sharp increase in the NADPH content [45]. The elevated levels of NADPH led to excessive reduction of the inter-photosystem electron transport components, including the cytochrome b/f complex and the PQ pool [43,45]. This effect hinders the electron transfer from PS II to PS I, as evidenced by the inhibited transfer of PSII electrons from QB to PQ pool and the reduced ETR_m_ in the present study (Figure 5c,d).

In NaAc culture, the mutant exhibited lower photosynthetic pigment content compared to the wild type, yet maintained the same photosynthetic efficiency (Figure 4 and Figure 5a–c). This may be associated with the rebalancing of photon conversion and photon utilization efficiency. Huang et al. [46] found that when the photosynthetic pigment content decreased, the cyanobacterium *Microcystis flosaquae* enhanced photosynthetic activity by stabilizing the actual photon conversion efficiency and photon utilization efficiency.

### 3.3. Lipid and Fatty Acids Analysis

Under NaAc culture conditions, both wild-type and mutant cells showed the presence of lipid droplets (golden fluorescence) in the late-exponential growth phase, while no apparent lipid droplets were observed under conditions without NaAc (Appendix A). This indicates that lipid accumulation occurred in subcellular structures of both wild-type and mutant strains under acetate assimilation conditions.

The crude lipid content after cultivation for seven days is shown in Figure 6a. In the absence of NaAc, the wild-type exhibited a crude lipid content of 0.26 g·g^−1^, while the mutant strain showed a crude lipid content of 0.34 g·g^−1^. When cultured with NaAc, there was a significant increase in the crude lipid content for both strains. Specifically, the wild-type strain exhibited an enhancement to 0.38 g·g^−1^, which corresponds to a 46.15% increase compared to WT-0M. Similarly, the mutant strain demonstrated an increase to 0.47 g·g^−1^, amounting to a 38.24% augmentation compared to ADY2-4-0M. This is consistent with the results of fluorescence observation, as mentioned above. The promoting effect of acetate on lipid accumulation has been widely reported in microalgae, such as *P. tricornutum* [12], *Navicula saprophila* [47], and *C. vulgaris* [41]. However, the study conducted by Haro et al. reported that acetate promoted growth but did not enhance lipid accumulation in a Chilean strain of *P. tricornutum* [32]. The deviation may be related to the intrinsic characteristics of the strain.

In the absence of NaAc, the total fatty acid content did not exhibit any significant variance (*p* > 0.05, Figure 6b). Nevertheless, upon the introduction of NaAc into the culture medium, a noteworthy discrepancy emerged between the two strains. Specifically, the total fatty acid content of the wild-type remained statistically unchanged (*p* > 0.05), whereas that of the mutant experienced a pronounced elevation. This increase was measured to be 42.35% relative to the levels observed in ADY2-4-0M.

The fatty acid compositions after 7 days of cultivation are illustrated in Table 3. The main components of fatty acids in each group include C14:0 (6.17–8.92%), C16:0 (11.97–23.64%), C16:1n-7 (16.00–33.82%), C18:1n-9 (6.58–9.41%), and EPA (12.59–27.19%). After treatment with NaAc, the percentage content of C16:0 and C16:1n-7 in both the wild-type and mutant strains significantly increased, while the percentage content of EPA decreased (*p* < 0.05). This trend is consistent with the findings reported by Haro et al. in *P. tricornutum* treated with acetate [32]. Compared to WT-0M, ADY2-4-0M increased the relative content of C16:0 and C16:1n-7 while decreasing the relative content of EPA. Compared to WT-0.01M, ADY2-4-0.01M increased the relative content of C16:1n-7 and slightly reduced the relative content of EPA, while maintaining the relative content of C16:0. The primary fatty acids synthesized in the chloroplast of *P. tricornutum* are C16:0 and C16:1n-7 [48], while EPA derived from C16:0 through C18:0 and C18:1n-9 are mainly synthesized in the endoplasmic reticulum [49,50]. Therefore, EPA and C16:1n-7 compete for the substrate C16:0, exhibiting a competitive relationship of trade-off [51]. The results of this study provide a good explanation of this mechanism. It is also indicated that the mutant enhanced the synthesis of C16:1n-7 by strengthening acetate transport and utilization, thereby strengthening the prokaryotic pathway within the chloroplast [52].

Without the addition of NaAc in the culture, the UI of fatty acids in the mutant strain was significantly lower than that of the wild-type (*p* < 0.05). However, under conditions of culturing with NaAc, there was no significant difference in the UI between the wild-type and the mutant strain, but it decreased by 35.51% and 34.22%, respectively, compared to when cultured without NaAc (*p* > 0.05). Previous reports have indicated a strong positive correlation between the polar/neutral lipids ratio and the UI in microalgae [53,54]. In this study, assimilation of acetate led to a decrease in the UI, which also implies a decrease in the percentage content of polar lipids. In general, polar lipids, which play a role in forming membrane structures, typically contain higher levels of PUFA compared to other lipids [55]. The reduction in PUFA (Table 3, EPA) in the polar lipids often predicts a potential decrease in membrane fluidity. Whether this decrease in fluidity benefits the conformational change of acetate transport protein to facilitate the transport of acetate thus remains to be investigated.

## 4. Conclusions

The introduction of the acetate transport protein ADY2 from *S. cerevisiae* into *P. tri-cornutum* successfully conferred the microalga with an enhanced capacity for acetate assimilation, leading to accelerated growth. The transformant ADY2-4 demonstrated significantly higher growth rates and increased dry weight under 0.01 M NaAc conditions compared to the wild-type. Additionally, the strain exhibited the ability to assimilate other short-chain fatty acids such as sodium propionate and sodium butyrate. Regarding photosynthesis, although pigment content decreased with NaAc treatment, the mutant strain displayed higher pigment content in the absence of NaAc compared to the wild type. In terms of lipid and fatty acid profiles, the addition of NaAc significantly increased the crude lipid content in the mutant, with a notable increase in total fatty acid content. The fatty acid composition analysis indicated an increase in C16:0 and C16:1n-7 content and a decrease in EPA content and the UI after NaAc treatment. These results suggest that the mutant strain bolsters the prokaryotic pathway within the chloroplast by enhancing acetate transport and utilization. The study presents a novel perspective on improving organic carbon source utilization in microalgae through metabolic engineering, offering potential strategies for the development of microalgae-based biofuels and bioproducts.

## Figures and Tables

**Figure 1 biomolecules-14-00822-f001:**
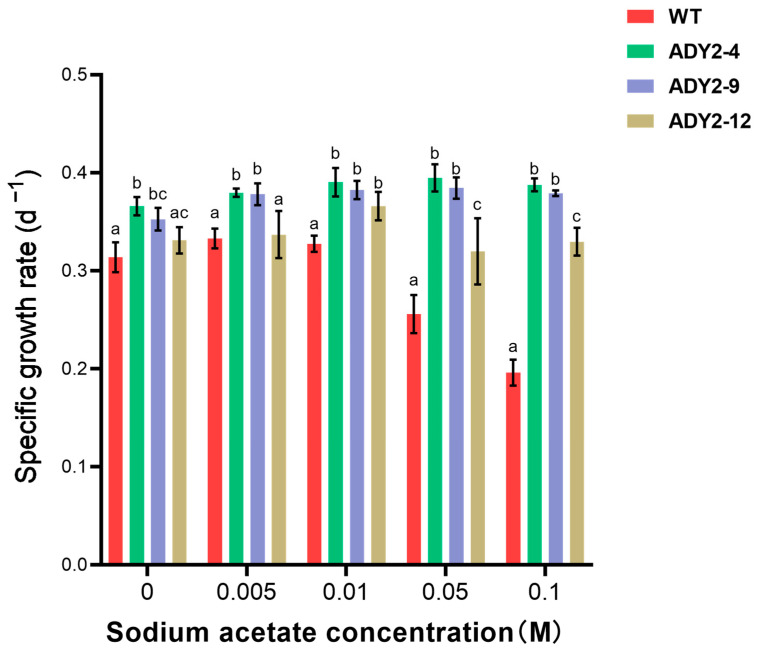
The specific growth rates of *P. tricornutum* cells under different concentrations of sodium acetate. Column values (mean ± SD of three replicates) with different lowercase letters are significantly different (*p* < 0.05).

**Figure 2 biomolecules-14-00822-f002:**
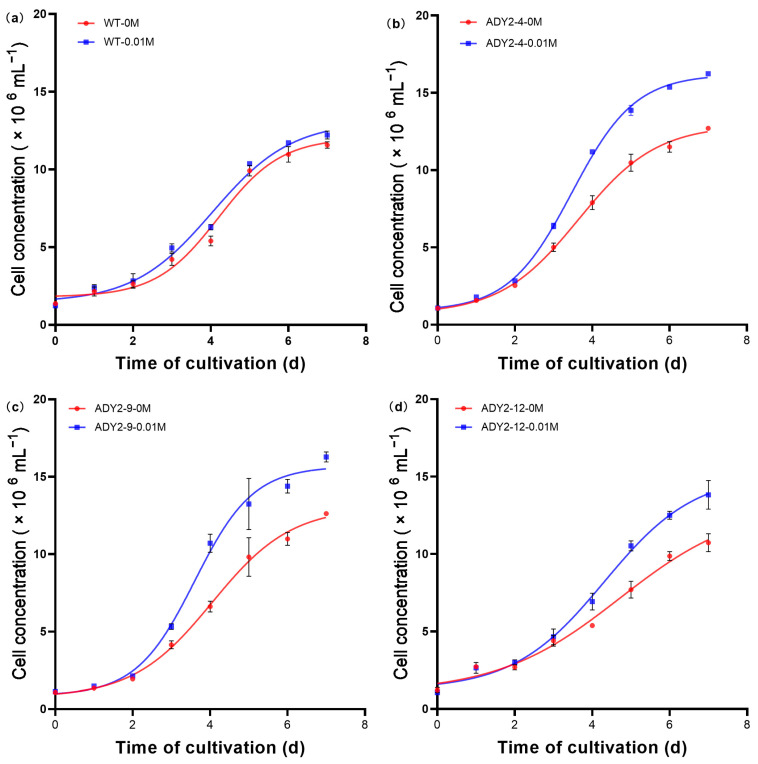
The growth curves of (**a**) WT, (**b**) ADY2-4, (**c**) ADY2-9, and (**d**) ADY2-12 strains cultivated under conditions with and without the addition of NaAc.

**Figure 3 biomolecules-14-00822-f003:**
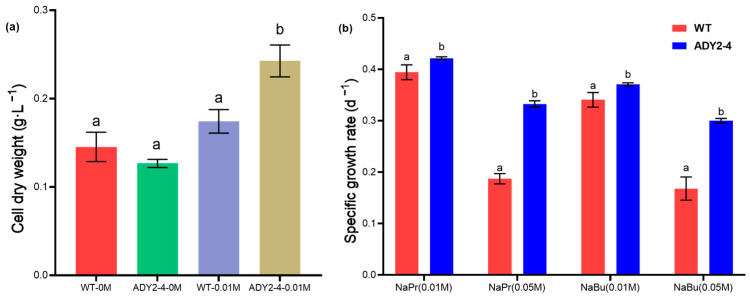
(**a**) Comparison of the cell dry weight at the end of the growth experiment for cells with or without 0.01 M NaAc, and (**b**) the specific growth rate of cells treated with 0.01 M or 0.05 M sodium propionate (NaPr) or sodium butyrate (NaBu). Column values (mean ± SD of three replicates) with different lowercase letters are significantly different (*p* < 0.05).

**Figure 4 biomolecules-14-00822-f004:**
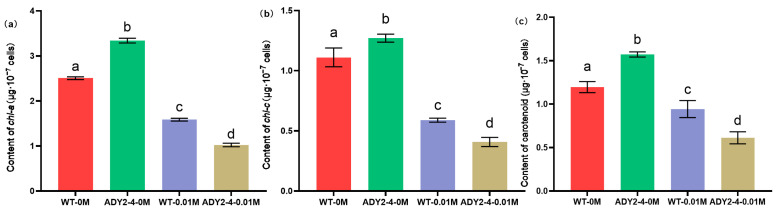
Photosynthetic pigments content per 10^7^ cells of the wild-type and ADY2-4 under different treatments. (**a**) The content of *Chl-a*; (**b**) the content of *Chl-c*; (**c**) the content of carotenoid. Column values (mean ± SD of three replicates) with different lowercase letters are significantly different (*p* < 0.05).

**Figure 5 biomolecules-14-00822-f005:**
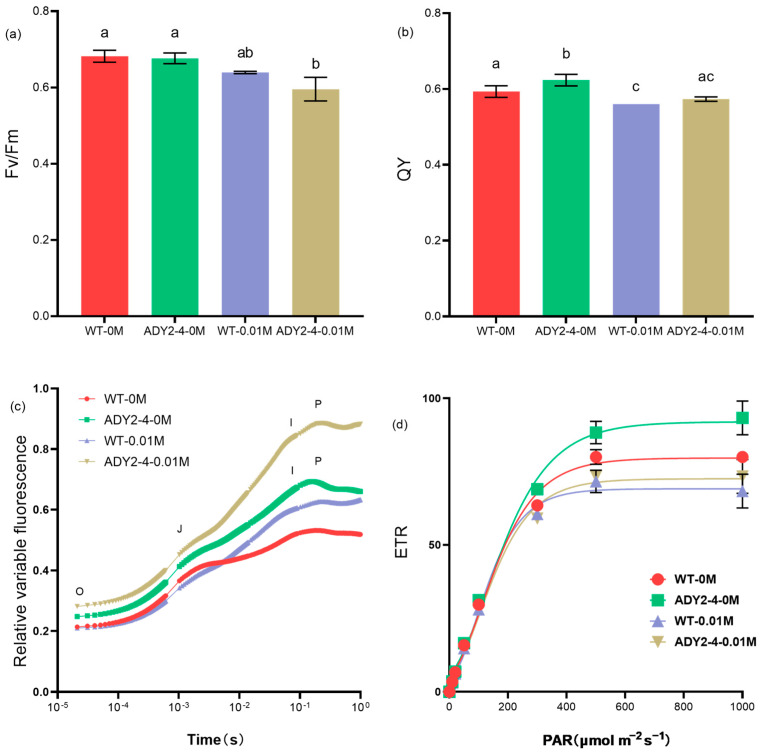
(**a**) Responses of maximum quantum yield of PSII (*F_v_*/*F_m_*), (**b**) the effective quantum yield (QY), (**c**) rapid chlorophyll fluorescence induction curves (OJIP-test), and (**d**) light response curves of the wild-type and mutant strains under conditions with and without the addition of NaAc. Column values (mean ± SD of three replicates) with different lowercase letters are significantly different (*p* < 0.05).

**Figure 6 biomolecules-14-00822-f006:**
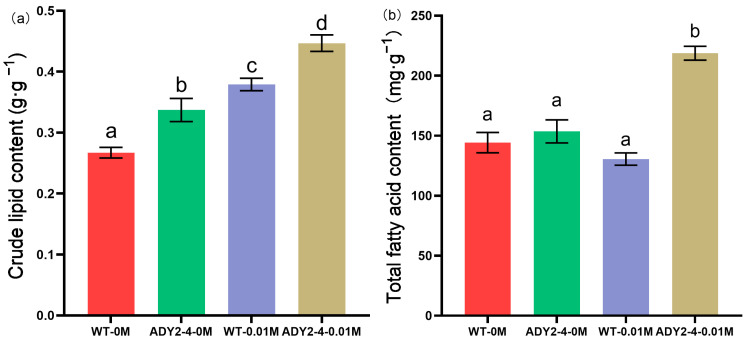
(**a**) The crude lipid content and (**b**) total fatty acid content of the wild-type and mutant strains under conditions with or without the addition of sodium acetate. Column values (mean ± SD of three replicates) with different lowercase letters are significantly different (*p* < 0.05).

**Table 1 biomolecules-14-00822-t001:** Acetate transporter gene *ADY2* primer sequences for PCR amplification.

Primer	Sequence (5′-3′)
ADY2-PCR-F	CATGCATGCTCTCGTTATTAGTAGGTCGTGC
ADY2-PCR-R	CGCGGATCCGTTGGATCGTATTTCCCTGAG

**Table 2 biomolecules-14-00822-t002:** The removal efficiency of COD by wild-type and mutant strains. Values (mean± SD of three replicates) with different lowercase letters are significantly different (*p* < 0.05).

Strains	Initial COD (mg L^−1^)	Final COD (mg L^−1^)	COD Removal Rate (%)
WT	180.58 ± 8.21	24.86 ± 1.23 ^a^	86.23 ± 0.25 ^a^
ADY2-4	180.58 ± 8.21	17.51 ± 1.25 ^b^	90.30 ± 0.27 ^b^

**Table 3 biomolecules-14-00822-t003:** The fatty acid composition (% of total fatty acids) and unsaturation index (UI) of the wild-type and mutant strains under conditions with or without the addition of sodium acetate. Values (mean ± SD of three replicates) within the same row with different lowercase letters are significantly different (*p* < 0.05).

Fatty Acids	Groups			
	WT-0M	ADY2-4-0M	WT-0.01M	ADY2-4-0.01M
C14:0	8.92 ± 0.28 a	7.69 ± 0.32 b	7.69 ± 0.12 b	6.17 ± 0.16 c
C16:0	11.97 ± 0.75 a	14.63 ± 0.29 b	23.64 ± 1.05 c	22.76 ± 0.37 c
C16:1n-7	16.01 ± 0.56 a	20.50 ± 0.54 b	28.17 ± 1.95 c	33.82 ± 0.76 d
C18:1n-9	9.33 ± 0.13 a	8.17 ± 0.20 b	9.41 ± 0.33 a	6.58 ± 0.12 c
EPA	27.19 ± 0.65 a	24.42 ± 0.25 b	13.41 ± 0.33 c	12.59 ± 0.45 c
C16:4	3.00 ± 0.25 a	3.53 ± 0.15 b	1.40 ± 0.10 c	1.25 ± 0.09 c
C18:0	0.56 ± 0.09 a	0.55 ± 0.02 a	1.44 ± 0.19 b	0.82 ± 0.03 a
C18:1n-7	0.55 ± 0.02 a	0.81 ± 0.03 b	1.19 ± 0.08 c	2.17 ± 0.14 d
C18:2n-6	2.91 ± 0.04 a	2.15 ± 0.18 b	1.22 ± 0.05 c	1.35 ± 0.06 c
C20:3n-3	1.79 ± 0.15 a	1.36 ± 0.02 b	1.02 ± 0.06 c	0.05 ± 0.00 d
C20:4n-3	1.06 ± 0.05 a	0.50 ± 0.11 b	0.33 ± 0.03 b	0.44 ± 0.02 b
C24:0	1.51 ± 0.11 a	1.53 ± 0.03 a	0.89 ± 0.06 b	0.84 ± 0.02 b
Minor FAs *	2.39 ± 0.04 a	2.52 ± 0.03 a	2.66 ± 0.40 a	1.81 ± 0.06 b
UI	200.59 ± 3.51 a	188.80 ± 0.92 b	129.36 ± 2.32 c	124.20 ± 2.73 c

* The sum of the percentages of the fatty acids including C16:2, C16:3, C18:3n-3, C18:3n-6, C20:4n-6, C22:5n-3, and DHA, with the proportion of each minor component ranging from 0.01% to 0.86%.

## Data Availability

All data generated or analyzed during this study are included in this published article and its Appendix A.

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
