# Peer review of "Enhancing Acetate Utilization in Phaeodactylum tricornutum through the Introduction of Acetate Transport Protein"

_biomolecules, 2024, doi:10.3390/biom14070822_

Round 1
Reviewer 1 Report
Comments and Suggestions for Authors
The article by Pu Song et al. is devoted to the undoubtedly urgent topic of biotechnology of microalgae that accumulate significant amounts of fatty oil in order to obtain nutraceutically valuable lipids and polyunsaturated fatty acids. The authors selected Phaeodactylum tricornutum, a strain of microalgae that possesses a number of unique features that make it attractive for biotechnological applications. The authors subjected cells of this algae to genetic transformation, thereby enabling them to transport exogenous acetate and incorporate it into their metabolism. As a consequence, the authors observed an increase in the "oiliness" of the transformants compared to wild-type cells, accompanied by a change in the fatty acid composition of total lipids, which was mainly expressed as a decrease in their unsaturation.
In general, the work is of a high standard, having been carried out at a high methodological and technical level using methods that have become common in studies of this kind worldwide. However, upon careful reading of the text, a number of remarks arose which I would like to draw the attention of the authors to.
Firstly, it is necessary to highlight the method employed to determine the content of total lipids. The authors utilised the gravimetric method of lipid determination, whereby the mass of the dry residue in a flask was determined following the evaporation of solvents from the extract obtained by Folch's method. It is important to note that this approach is a highly simplified methodology. It is inevitable that the extract, in addition to acyl-containing lipids, will also contain substances of non-lipid nature, which are lipophillic. These can be pigment molecules, free sterols, secondary metabolism compounds, peptides, and others. Consequently, it is inaccurate to categorise such an extract as "lipids".
Conversely, the authors employed the methodology of fatty acid analysis, incorporating an internal standard. If the mass or number of cells in each sample and the quantity of standard added are known, the content of total lipids may be accurately determined through the amount of fatty acids acylating them. It is recommended that the authors recalculate the total lipid content in this manner and correct the data in Figure 5 by expressing it as mg/g dry weight of fatty acids. In this case, section 2.7 can be omitted, as direct esterification, as employed by the authors, provides a more comprehensive understanding of the quantity of fatty acids and, consequently, the lipids acylated by them than extraction.
It is necessary to highlight that the method of obtaining methyl esters is not sufficiently detailed. The authors refer to their 2016 publication, which in turn refers to the article by Griffiths, van Hille, and Harrison (2010). It is recommended that the authors provide a detailed description of the method used in the manuscript rather than merely citing previous publications.
In the Results section, the authors demonstrate that cells transformed in vitro and cultured in medium containing exogenous acetate exhibit an increase in lipid content, while chlorophylls decrease. It can be observed that there are alterations in the ultrastructure of the cells, a reduction in the number of plastids, the accumulation of starch grains and oleosomes. It would be beneficial to include images of the cell ultrastructures and to discuss the observed changes in comparison with those of the wild-type cells. In addition, there is a reduction in the proportion of polyunsaturated LCs, which are typically found in microalgae as part of the glycolipids of photosynthetic membranes rather than in triacylglycerols.
Figure 6, which illustrates the alterations in fatty acid composition across experimental variants, is challenging to interpret. It is recommended that the data be presented in tabular format.
Furthermore, it would be highly beneficial to supplement the data on total lipids with the absolute and relative content of triacylglycerols in cells by experiment variants and their fatty acid composition.
Author Response
Reviewer1
The article by Pu Song et al. is devoted to the undoubtedly urgent topic of biotechnology of microalgae that accumulate significant amounts of fatty oil in order to obtain nutraceutically valuable lipids and polyunsaturated fatty acids. The authors selected Phaeodactylum tricornutum, a strain of microalgae that possesses a number of unique features that make it attractive for biotechnological applications. The authors subjected cells of this algae to genetic transformation, thereby enabling them to transport exogenous acetate and incorporate it into their metabolism. As a consequence, the authors observed an increase in the "oiliness" of the transformants compared to wild-type cells, accompanied by a change in the fatty acid composition of total lipids, which was mainly expressed as a decrease in their unsaturation.
In general, the work is of a high standard, having been carried out at a high methodological and technical level using methods that have become common in studies of this kind worldwide. However, upon careful reading of the text, a number of remarks arose which I would like to draw the attention of the authors to.
Answer: We very much appreciate the careful reading of our manuscript and valuable suggestions of the reviewer. We have carefully considered the comments and have revised the manuscript accordingly. Revised portion are marked in the paper with red text.
Q1: Firstly, it is necessary to highlight the method employed to determine the content of total lipids. The authors utilised the gravimetric method of lipid determination, whereby the mass of the dry residue in a flask was determined following the evaporation of solvents from the extract obtained by Folch's method. It is important to note that this approach is a highly simplified methodology. It is inevitable that the extract, in addition to acyl-containing lipids, will also contain substances of non-lipid nature, which are lipophillic. These can be pigment molecules, free sterols, secondary metabolism compounds, peptides, and others. Consequently, it is inaccurate to categorise such an extract as "lipids".
Answer: We are grateful for the suggestion. Folch's method is a commonly used lipid extraction method in microalgae. It has been widely reported in the literatures, such as Feng et al., 2013, Liu et al., 2011 and Figueiredo et al, 2019. Indeed, as you mentioned, the lipids here include other components such as pigments. To be more clear and in accordance with your concerns, we have modified it to the content of crude lipid, in order to address your question. Please see details at P11, line390-402, and Figure 6.
Feng, G. D., Zhang, F., Cheng, L. H., Xu, X. H., Zhang, L., & Chen, H. L. (2013). Evaluation of FT-IR and Nile Red methods for microalgal lipid characterization and biomass composition determination. Bioresource technology, 128, 107-112.
Liu, J., Huang, J., Sun, Z., Zhong, Y., Jiang, Y., & Chen, F. (2011). Differential lipid and fatty acid profiles of photoautotrophic and heterotrophic Chlorella zofingiensis: assessment of algal oils for biodiesel production. Bioresource technology, 102(1), 106-110.
Figueiredo, A. R., da Costa, E., Silva, J., Domingues, M. R., & Domingues, P. (2019). The effects of different extraction methods of lipids from Nannochloropsis oceanica on the contents of omega-3 fatty acids. Algal research, 41, 101556.
Q2: Conversely, the authors employed the methodology of fatty acid analysis, incorporating an internal standard. If the mass or number of cells in each sample and the quantity of standard added are known, the content of total lipids may be accurately determined through the amount of fatty acids acylating them. It is recommended that the authors recalculate the total lipid content in this manner and correct the data in Figure 5 by expressing it as mg/g dry weight of fatty acids. In this case, section 2.7 can be omitted, as direct esterification, as employed by the authors, provides a more comprehensive understanding of the quantity of fatty acids and, consequently, the lipids acylated by them than extraction.
Answer: We are grateful for the suggestion. The fatty acids determined by the method we used are the products of methylation. However, microalgae contain triglycerides, phospholipids, glycolipids, and other forms in P. tricornutum. Therefore, the content of fatty acid methyl esters cannot represent the total lipid content. Here we used the internal standard method to determine the total fatty acid content. By comparing the differences in total fatty acid content between the mutant and wild-type, we obtained the conclusion that the acetate transporter protein can promote fatty acid accumulation. Your question highlights the significance of studying the impact of the mutant on different types of lipids such as phospholipids, glycolipids, and triglycerides. We will conduct more detailed studies in the future.
Q3: It is necessary to highlight that the method of obtaining methyl esters is not sufficiently detailed. The authors refer to their 2016 publication, which in turn refers to the article by Griffiths, van Hille, and Harrison (2010). It is recommended that the authors provide a detailed description of the method used in the manuscript rather than merely citing previous publications.
Answer: Thanks a lot for your suggestion. Our transesterification method is described as follows: Typically, an appropriate amount of wet cell pellet (c.a. 100 mg) was accurately weighed into a screw-capped test tube (12 cm X 1.5 cm). Half milliliter of toluene containing internal standard (0.1 mg glyceryl triheptadecanoate) and 1 mL of NaOH methanol solution were slowly added and vortex-mixed at 80°C with constant stirring at 300 rpm for 20 min. The mixture is cooled to room temperature for 5 min, followed by addition of 1 mL of hydrochloric acid methanol solution, repeating the above incubation procedure. After cooling for 5 min to room temperature, 1 mL of 6% K2CO3 and 400 μL of n-hexane are added, vortex-mixed, and the supernatant was extracted by centrifugation at 4000 rpm for 1 min, followed by collection of the upper layer for gas chromatography analysis. P5, line198-213.
Q4: In the Results section, the authors demonstrate that cells transformed in vitro and cultured in medium containing exogenous acetate exhibit an increase in lipid content, while chlorophylls decrease. It can be observed that there are alterations in the ultrastructure of the cells, a reduction in the number of plastids, the accumulation of starch grains and oleosomes. It would be beneficial to include images of the cell ultrastructures and to discuss the observed changes in comparison with those of the wild-type cells.
Answer: Thanks a lot for your suggestion. We provided images of lipid droplets stained with Nile Red fluorescent dye for observation. Under NaAc culture conditions, both wild-type and mutant cells showed the presence of lipid droplets (golden fluorescence) in the late-exponential growth phase, while no apparent lipid droplets were observed under conditions without NaAc (Figure S4). This indicates that lipid accumulation occurred in subcellular structures of both wild-type and mutant strains under acetate assimilation conditions. P4, line175-182; P11, line385-389; Figure S4.
Q5: In addition, there is a reduction in the proportion of polyunsaturated LCs, which are typically found in microalgae as part of the glycolipids of photosynthetic membranes rather than in triacylglycerols. Figure 6, which illustrates the alterations in fatty acid composition across experimental variants, is challenging to interpret. It is recommended that the data be presented in tabular format.
Answer: Revise accordingly. Figure 6 has been presented in tabular format.
Q6: Furthermore, it would be highly beneficial to supplement the data on total lipids with the absolute and relative content of triacylglycerols in cells by experiment variants and their fatty acid composition.
Answer: This is truly an excellent suggestion. The proportion of triacylglycerols, phospholipids, and glycolipids in total lipid plays a crucial role in gaining a comprehensive understanding of the types of accumulated lipid in the mutant strain, which will be another important research direction. We will conduct detailed research on this in the future.
Reviewer 2 Report
Comments and Suggestions for Authors
The Authors show the phenotype of a diatom specie tranformation wiht an acetate transporter. The work is well done and fluent. I just have few remarks as follow:
- Line 87 and 88: don't use abbreviation when they appear for the first time but write it by extensio (YPD, ADY2)
-Line 91: add "with primers listed in table 1"; fcpA means?
-Line 92: pPha-T1, is htere any reference for this?
-Line 94: TOP10 cells, add trade mark;
-Line 117: "on the dry weight content of the strain", which strain?
-Line 116: "After selecting the rapid growth strain, a comparative analysis was conducted on the dry weight content of the strain and the wild-type strain after 7 days of growth under 0M and 0.01M NaAc conditions", while only these two concentrations?
-Line 131: "was calculated using formula (1)." , substitute with : "was calculated using the following formula: put the formula here;
-Line137: "onto a pretreated glass fiber filter", pretreated with?
-Line 140 on: you passed from a discorsive description to a list type description of the method steps, why?
-Paragraph 2.7: the paragraph is incomplete, please complete it;
-Figure 1b: use different color for each curve and use bigger simbol. It is difficult to distringuish among the different clones/treatments. In addition, why you did not showed the curve for each treatment?
in figure legend in general, the lowercase letters which comparison significance indicate? who is the referring point/tratment?
Author Response
Reviewer2
The Authors show the phenotype of a diatom specie tranformation wiht an acetate transporter. The work is well done and fluent. I just have few remarks as follow:
Answer: We thank you so much for your constructive suggestions to help us improve the manuscript. We have carefully considered the comments and have revised the manuscript accordingly. Revised portion are marked in the paper with red text.
- Line 87 and 88: don't use abbreviation when they appear for the first time but write it by extensio (YPD, ADY2)
Answer: Revise accordingly. YPD is the abbreviation of the yeast extracts-peptone-dextrose medium. ADY2 is a gene name of the yeast Saccharomyces cerevisiae, and we do not find the full name. P2, line87-88.
-Line 91: add "with primers listed in table 1"; fcpA means?
Answer: Revise accordingly. FcpA is the name of the gene promoter on the plasmid vector. P2, line 92.
-Line 92: pPha-T1, is htere any reference for this?
Answer: We added the original reference of pPha-T1. P2, line94; P14, line517-518.
Zaslavskaia, L. A., Lippmeier, J. C., Kroth, P. G., Grossman, A. R., & Apt, K. E. (2000). Transformation of the diatom Phaeodactylum tricornutum (Bacillariophyceae) with a variety of selectable marker and reporter genes. Journal of phycology, 36(2), 379-386.
-Line 94: TOP10 cells, add trade mark;
Answer: Revise accordingly. TOP10 (Sangon Biotech. Co., Ltd, Shanghai, China). P95.
-Line 117: "on the dry weight content of the strain", which strain?
Answer: Revise accordingly. The mutant strain. Line120.
-Line 116: "After selecting the rapid growth strain, a comparative analysis was conducted on the dry weight content of the strain and the wild-type strain after 7 days of growth under 0M and 0.01M NaAc conditions", while only these two concentrations?
Answer: In the preceding analyses, we have examined the cell growth conditions at five concentrations: 0M, 0.005M, 0.01M, 0.05M, and 0.1M. The concentration of 0.01M, which exhibited the best growth conditions, was selected for further analysis.
-Line 131: "was calculated using formula (1)." , substitute with : "was calculated using the following formula: put the formula here;
Answer: Revise accordingly. Line133-137.
-Line137: "onto a pretreated glass fiber filter", pretreated with?
Answer: An accurate volume of the culture was filtered onto a pre-dried and pre-weighed glass fiber filter. Line 139.
-Line 140 on: you passed from a discorsive description to a list type description of the method steps, why?
Answer: The language we used may have been confusing. We have rephrased the following method steps, hoping to address your concerns. Both filters were dried at 100°C for 4 hours to allow the ammonium formate to evaporate, and then weighed on an analytical balance, respectively. Line142-144.
-Paragraph 2.7: the paragraph is incomplete, please complete it;
Answer: We apologize for making this error while preparing the manuscript, and we have now supplemented the method for total lipid determination. Line176-183.
-Figure 1b: use different color for each curve and use bigger simbol. It is difficult to distringuish among the different clones/treatments. In addition, why you did not showed the curve for each treatment?
Answer: Revise accordingly. Figure 1 is divided into two separate figures: one displaying the specific growth rates, while the other shows the growth curves of the wild-type and mutant strains individually. Please see details at figure 1 and figure 2. Line 270-276.
in figure legend in general, the lowercase letters which comparison significance indicate? who is the referring point/tratment?
Answer: We apologize for not explaining the statistical methods clearly. In the text, we used two different statistical methods. Data with parametric distribution were subjected to one-way analysis of variance (more than two groups) or t-test (two groups). As for figure3b, t-test was used for significance test analysis, and one-way analysis of variance was used in the other figures. Line226-227.
Reviewer 3 Report
Comments and Suggestions for Authors
In this document the authors describe the introduction of the acetate transporter gene ADY2 from Saccharomyces cerevisiae into the diatom Phaeodactylum tricornutum, resulting in an increased capability to assimilate acetate and accelerating its growth which improves the total lipid content and modify the fatty acid composition, at an optimal NaAc concentration of 0.01 M. A minor suggestion which may improve the quality of the document is the addition of some brief description in the Introduction, of the pPha-Ti plasmid and characteristics of the fcpA promoter used, mainly for readers without background in the diatom system. Minor corrections in lines 175 and 179, change uL by microL.
Author Response
Reviewer3
In this document the authors describe the introduction of the acetate transporter gene ADY2 from Saccharomyces cerevisiae into the diatom Phaeodactylum tricornutum, resulting in an increased capability to assimilate acetate and accelerating its growth which improves the total lipid content and modify the fatty acid composition, at an optimal NaAc concentration of 0.01 M. A minor suggestion which may improve the quality of the document is the addition of some brief description in the Introduction, of the pPha-Ti plasmid and characteristics of the fcpA promoter used, mainly for readers without background in the diatom system. Minor corrections in lines 175 and 179, change uL by microL.
Answer: Thanks a lot for suggestion. We added the original reference of the pPha-Ti plasmid in the text. The supplementary materials also provide the plasmid maps, hoping to address your concerns. Minor corrections of uL have also been revised. Please see details at line 93 and line 186, 190. Revised portion are marked in the paper with red text.
Zaslavskaia, L. A., Lippmeier, J. C., Kroth, P. G., Grossman, A. R., & Apt, K. E. (2000). Transformation of the diatom Phaeodactylum tricornutum (Bacillariophyceae) with a variety of selectable marker and reporter genes. Journal of phycology, 36(2), 379-386.
Round 2
Reviewer 1 Report
Comments and Suggestions for Authors
Firstly, I would like to express my gratitude to the authors for their efforts to enhance the format of presentation of their results and the manuscript.
Upon meticulous examination of the responses to the previous critiques, it is evident that the authors are cognizant of the necessity to further investigate the transformant of the diatom Phaeodactylum tricornutum, with the objective of elucidating in subsequent studies the influence of endogenous sodium acetate on glycerolipid metabolism in the transformants' cells. These works will undoubtedly contribute to a deeper understanding of the regulatory pathways involved in glycerolipid metabolism in such organisms, with the potential to produce lipids with defined nutraceutical properties. While I do not concur with all the responses, I believe that the paper can be published in its current form with a number of minor clarifications.
First of all, it should be clarified why the described method for the preparation of fatty acid methyl esters of total cellular lipids does not coincide with the method referred to by the authors. The point is that in line 199 they write:
«The fatty acid composition was analyzed according to our prior publication [27]».
Reference 27 is their 2016 article entitled "Effect of culture conditions on growth, fatty acid composition and DHA/EPA ratio of Phaeodactylum tricornutum". This article states the following: "Direct transesterification was used for the fatty acid analysis according to Griffiths et al. (2010).
The transesterification procedure described in Griffiths et al (2010) is as follows: «A combination of base followed by acid catalysis was performed as follows: samples were dissolved in 500 ul toluene containing 0.1 mg C17-TAG in glass test tubes with silicon-lined screw-cap lids. One hundred microliters of 2,2-dimethoxypropane (water scavenger) was added. Sodium methoxide (1 ml) was then added and the samples mixed briefly by vortexing before being placed in an incubator at 80 °C, with shaking at 300 rpm for 20 min. Samples were cooled for 5 min to room temperature and 1 ml BF 3 methanol was added before repeating the incubation. After cooling for 5 min to room temperature, 400 ul dH2O and 400 ul hexane containing 0.1 mg C19-ME were added and tubes mixed by vortexing. Samples were centrifuged at 4,000 rpm for 1 min and the upper hexane-toluene layer, containing the FAME extract, was transferred to vials for GC».
It is clear that the protocol description added by the authors (section 2.9) does not match what they write about and refer to. This discrepancy should be clarified and corrected. I hope that this discrepancy does not deliberately mislead reviewers and readers about the methodology used by the authors.
Line 202: The concentration of KOH should be reported
Line 204: give the mass fraction of hydrochloric acid in methanol
Table 3: for ease of reading, all values should be rounded to tenths and the table should be shortened, leaving only those fatty acids that make up more than 1% of the total. All the others should be grouped together in the "minor FAs" line and listed in the footnote to the table as belonging to this group of FAs with a note that the proportion of each minor component was in the range 0.03-0.86%.
It is also very useful to calculate the unsaturation index (UI) per experimental variant and to add its discussion to the results.
UI = (1 × (% monoenoic) + 2 × (% dienoic) + 3 × (% trienoic) + 4 × (% tetraenoic) + 5 × (% pentaenoic) + 6 × (% hexaenoic)) / 100
This is a very sensitive and informative integral index characterising the unsaturation of fatty acids.
Author Response
Reviewer1
Comments1: Firstly, I would like to express my gratitude to the authors for their efforts to enhance the format of presentation of their results and the manuscript.
Upon meticulous examination of the responses to the previous critiques, it is evident that the authors are cognizant of the necessity to further investigate the transformant of the diatom Phaeodactylum tricornutum, with the objective of elucidating in subsequent studies the influence of endogenous sodium acetate on glycerolipid metabolism in the transformants' cells. These works will undoubtedly contribute to a deeper understanding of the regulatory pathways involved in glycerolipid metabolism in such organisms, with the potential to produce lipids with defined nutraceutical properties. While I do not concur with all the responses, I believe that the paper can be published in its current form with a number of minor clarifications.
Response1: We very much appreciate the careful reading of our manuscript and valuable suggestions of the reviewer. Your suggestions are constructive and inspiring, and have greatly contributed to our paper's improvement. We have carefully considered the comments and have revised the manuscript accordingly. Revised portion are marked in the paper with red text.
Comments2: First of all, it should be clarified why the described method for the preparation of fatty acid methyl esters of total cellular lipids does not coincide with the method referred to by the authors. The point is that in line 199 they write:
«The fatty acid composition was analyzed according to our prior publication [27]».
Reference 27 is their 2016 article entitled "Effect of culture conditions on growth, fatty acid composition and DHA/EPA ratio of Phaeodactylum tricornutum". This article states the following: "Direct transesterification was used for the fatty acid analysis according to Griffiths et al. (2010).
The transesterification procedure described in Griffiths et al (2010) is as follows: «A combination of base followed by acid catalysis was performed as follows: samples were dissolved in 500 ul toluene containing 0.1 mg C17-TAG in glass test tubes with silicon-lined screw-cap lids. One hundred microliters of 2,2-dimethoxypropane (water scavenger) was added. Sodium methoxide (1 ml) was then added and the samples mixed briefly by vortexing before being placed in an incubator at 80 °C, with shaking at 300 rpm for 20 min. Samples were cooled for 5 min to room temperature and 1 ml BF 3 methanol was added before repeating the incubation. After cooling for 5 min to room temperature, 400 ul dH2O and 400 ul hexane containing 0.1 mg C19-ME were added and tubes mixed by vortexing. Samples were centrifuged at 4,000 rpm for 1 min and the upper hexane-toluene layer, containing the FAME extract, was transferred to vials for GC».
It is clear that the protocol description added by the authors (section 2.9) does not match what they write about and refer to. This discrepancy should be clarified and corrected. I hope that this discrepancy does not deliberately mislead reviewers and readers about the methodology used by the authors.
Response2: We sincerely apologize for referencing an inappropriate paper. In fact, we published a paper in 2015 on the method for determining the fatty acid content of Phaeodactylum tricornutum, which is the method described in the present study. We have now replaced it with the 2015 paper. Thank you once again for your careful observation. P5, line552-553.
Qiao, H.; Wang, J.; Zhang, L.; Sun, C.; Ma, J.; Song, Z.; Li, B. An improved direct transesterification method for fatty acid determination of Phaeodactylum tricornutum. J. Appl. Phycol. 2015, 27, 697-701.
Comments3: Line 202: The concentration of KOH should be reported
Response2: revise accordingly. The concentration of NaOH is 0.5 M in anhydrous methanol. P5, line202.
Comments4: Line 204: give the mass fraction of hydrochloric acid in methanol
Response2: revise accordingly. The solution is 10:100 (v/v) acetyl chloride in anhydrous methanol. The acetyl chloride is used to produce hydrochloric acid. P5, line205.
Comments5: Table 3: for ease of reading, all values should be rounded to tenths and the table should be shortened, leaving only those fatty acids that make up more than 1% of the total. All the others should be grouped together in the "minor FAs" line and listed in the footnote to the table as belonging to this group of FAs with a note that the proportion of each minor component was in the range 0.03-0.86%.
Response5: revise accordingly. P12, Table 3.
Comments6: It is also very useful to calculate the unsaturation index (UI) per experimental variant and to add its discussion to the results.
UI = (1 × (% monoenoic) + 2 × (% dienoic) + 3 × (% trienoic) + 4 × (% tetraenoic) + 5 × (% pentaenoic) + 6 × (% hexaenoic)) / 100
This is a very sensitive and informative integral index characterising the unsaturation of fatty acids.
Response6: revise accordingly. The unsaturation index (UI) was introduced in the abstract, methods, table3, results and discussions (P1, line24; P5, line214-217; P12, line4.34-446; P13, line465). We gave the description of the results and discussed the results as follows:
Without the addition of NaAc in the culture, the UI of fatty acids in the mutant strain was significantly lower than that of the wild-type (P < 0.05). However, under conditions of culturing with NaAc, there was no significant difference in the UI between the wild-type and the mutant strain, but it decreased by 35.51% and 34.22%, respectively, compared to when cultured without NaAc (P > 0.05). Previous reports have indicated a strong positive correlation between the polar/neutral lipids ratio and the UI in microalgae [1,2]. In this study, assimilation of acetate led to a decrease in UI, which also implies a decrease in the content of polar lipids. In general, polar lipids playing a role in forming membrane structures, typically contain higher levels of PUFA compared to other lipids [3]. The reducing PUFA (Table3, EPA) in the polar lipids often predicts a potential decrease in membrane fluidity. Whether this decrease in fluidity benefits the conformational change of acetate transport protein to facilitate the transport of acetate thus remains to be investigated.
1.Mendoza Guzmán, H., de la Jara Valido, A., Carmona Duarte, L., & Freijanes Presmanes, K. (2011). Analysis of interspecific variation in relative fatty acid composition: use of flow cytometry to estimate unsaturation index and relative polyunsaturated fatty acid content in microalgae. Journal of Applied Phycology, 23, 7-15.
2.Mendoza Guzmán, H., de la Jara Valido, A., Freijanes Presmanes, K., & Carmona Duarte, L. (2012). Quick estimation of intraspecific variation of fatty acid composition in Dunaliella salina using flow cytometry and Nile Red. Journal of applied phycology, 24, 1237-1243.
3.Roessler PG (1990) Environmental control of glycerolipids metabolism in microalgae: commercial implications and future research directions. J Phycol 26:393–399.
